# FAKER: Generating Frequency-based Artificial Attributes via Random Walks for Non-attributed Graph Representation Learning

## Abstract

A key challenge for Graph Neural Networks (GNNs) is their reliance on initial node features, while many real-world graphs lack such attributes due to privacy constraints or limitations in data collection. Existing adjacency-only approaches attempt to learn representations directly from topology. However, they often inherit the sampling biases of random walks, leading to skewed embeddings. To address these limitations, we propose FAKER, a diagnosis-driven, model-agnostic framework that synthesizes artificial node attributes from topology alone. FAKER first analyzes group-level visit signals from random walks with Power Spectral Density (PSD) to quantify low-frequency persistence bias and high-frequency switching bias. The resulting quantified scores then drive an adaptive sampler that produces a balanced corpus without distributional assumptions by reweighting transitions and allocating additional walks. A lightweight co-occurrence encoder trained on this corpus yields dynamic features, which are merged with a compact structural summary and standardized to form plug-and-play attributes for any GNN. Across four benchmarks, FAKER establishes state-of-the-art results among adjacency-only baselines for node classification and link prediction. It also matches or outperforms feature-using methods on three datasets. Ablation and robustness studies show that the improvements result from the frequency-domain diagnosis and adaptive allocation, rather than from the number of walks or sensitive hyperparameter tuning. The code is available at:`https://anonymous.4open.science/r/FAKER-B41C`.

## 1 Introduction

When attributes are limited or missing, the appropriate inputs for a Graph Neural Network (GNN) become unclear. Many methods assume fully specified node features (Wang et al., 2024; Chen et al., 2020; Tu et al., 2025). In practice that assumption fails to hold in many cases because of privacy constraints, governance constraints, costly large-scale curation, withheld sensitive data (Xia et al., 2024; Li et al., 2023; Tu et al., 2024). This mismatch materially limits the practical deployment of vanilla GNN architectures (Figure 1) (Cui et al., 2022; Chen et al., 2020; Tu et al., 2025).

Under partial absence of node attributes, a large body of work reconstructs unobserved attributes from observed ones (Chen et al., 2020; Um et al., 2023; Tu et al., 2025; Jin et al., 2022; Peng et al., 2024; Xia et al., 2024; Tu et al., 2024). For example, SAT matches the distributions of attribute and structural embeddings to recover missing features (Chen et al., 2020). PCFI assigns channel-wise confidence so that reliable channels refine uncertain ones (Um et al., 2023). These approaches typically assume that researchers observe a sufficient fraction of attributes. As the missing rate increases, their performance degrades markedly (Rossi et al., 2022; Um et al., 2023).

In fully non-attributed graphs, structure-only synthesis is common (Cui et al., 2022). The same pipeline then trains skip-gram embeddings (Mikolov et al., 2013) and feeds them into a GNN (Cui et al., 2022). RAHG fuses multiple embeddings, such as Node2Vec and GraphWave (Donnat et al., 2018), to capture both proximity and structural roles (Li et al., 2023). Separately, Residual2Vec reduces degree-induced bias via a residual formulation relative to random-graph null models (Kojaku et al., 2021). Despite this progress, two issues persist in non-attributed graphs. First, random walks

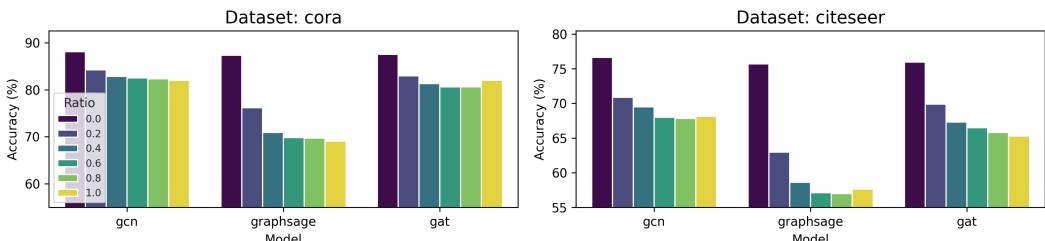

Figure 1: Comparison of GNN model accuracy under different attribute-missing ratios. A specified fraction of feature dimensions for every node is replaced with random noise.

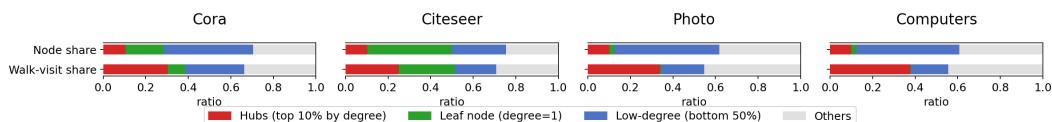

Figure 2: Distribution of random-walk visits across four graphs. Bars compare the node ratio with the corresponding visit ratio for each structural group. Settings: walk length = 20, walks per node = 40.

tend to over-visit hubs and under-visit low-degree nodes, which skews co-occurrences and hurts generalization (Kojaku et al., 2021; Rahman et al., 2019; Masuda et al., 2017). Second, the pipeline defers quantitative diagnosis of sampling bias until after embedding learning. That design choice, in turn, drives dataset-specific tuning of random-walk hyperparameters (Wu et al., 2020).

A standard approach for non-attributed graphs is to generate random walk based embeddings as the initial inputs to a GNN (Cui et al., 2022). However, random walks exhibit degree bias by oversampling hubs while undersampling low-degree nodes This bias produces hub-centric, local co-occurrence statistics (Figure 2). Message passing captures local topological patterns by design, so features derived from biased random walks impose redundant information on the GNN. As a result, hub representations grow disproportionately whereas low-degree representations diminish, which undermines generalization.

In this paper, we propose FAKER, a diagnosis-driven adaptive sampling framework. The core idea is to synthesize artificial features that complement a GNN's local aggregation. (i) Power Spectral Density (PSD) diagnosis quantifies, for each group, the low-frequency power that reflects node persistence and the high-frequency power that reflects group switching in visit signals. PSD separates these two biases into concise and comparable statistics, even from short sequences. These statistics map directly to the control variables of the walk strategy. (ii) Adaptive correction sets group-specific exploration parameters and allocates additional walks. It yields a balanced corpus that mitigates hub over-representation and low-degree under-representation. (iii) Feature synthesis trains an embedding model on the balanced corpus to obtain dynamic embeddings. It fuses these embeddings with a static Structural Identity (SI) vector, with the combined matrix standardized columnwise to produce the final features for the GNN.

The frequency-based design of FAKER offers three key advantages. First, the synthesized attributes are model-agnostic drop-in features that work with any backbone GNN. Second, the method tackles sampling bias at its source. The diagnosis of bias occurs during walk generation. The correction takes place before embedding training. This process yields representations that are stable as well as informative. Third, across diverse benchmarks FAKER consistently surpasses strong adjacency-only baselines. It also remains competitive with methods that use partial attributes. Controlled ablations and robustness checks trace these gains to the PSD-guided diagnosis and control rather than to walk volume or aggressive hyperparameter tuning.

## 2 RELATED WORK

**Graph Completion Learning**. Recent work in graph machine learning extensively studies imputing missing node attributes on partially attributed graphs. For instance, GCMmf (Taguchi et al., 2021) approximates the distribution of latent activations with a Gaussian Mixture Model (GMM), yielding more informative imputations. SAT (Chen et al., 2020) applies distribution matching in a shared latent space to recover missing attributes from structural signals. SVGA (Yoo et al., 2022) employs structured variational inference with a Gaussian Markov random field to model dependencies among latent variables. PCFI (Um et al., 2023) introduces a channel-wise confidence for each imputed attribute, computed via pseudo-confidence derived from the shortest path distance to the nearest node with observed attributes. WAGE (Tu et al., 2025) adopts a weight distribution encoder that tightly couples structure and attributes for reliable reconstruction under missingness. Amer (Jin et al., 2022) unifies attribute completion and representation learning rather than decoupling them. It maximizes mutual information to complete attributes, while an adversarial generative objective enforces structure attribute consistency. MATE (Peng et al., 2024) proposes a Dual Consistency Strategy (DCS) that jointly optimizes input space attributes and latent space representations by enforcing view wise consistency between structural and attribute information. AIAE (Xia et al., 2024) addresses noise and limited expressiveness in graph autoencoder based completion with a dual encoder design and knowledge distillation. This approach effectively fuses structural and attribute cues. RITR (Tu et al., 2024) tackles mixed missingness in which the cases of missing and incomplete attributes coexist by adopting an initialize-then-refine framework with tailored strategies for each type. The approaches above typically assume that a non trivial subset of attributes is observable, which restricts their applicability to non-attributed graphs where nodes start with no attributes at all. In contrast, our method FAKER does not attempt to reconstruct unobserved attributes. Instead, it synthesizes informative artificial attributes from scratch. Because generation is decoupled from GNN training, FAKER can be plugged into a wide range of GNN architectures without modifying their layers.

**Graph Embedding Methods for Non-attributed Graphs**. Learning on non-attributed graphs remains challenging because there is a lack of observed attributes for models to rely on. A common remedy is to synthesize node attributes via graph embeddings. DeepWalk (Perozzi et al., 2014) and node2vec (Grover & Leskovec, 2016) learn node embeddings by combining random walks with the skip-gram objective, with node2vec introducing controllable BFS/DFS trade-offs via $p$ and $q$. VERSE (Tsitsulin et al., 2018) learns embeddings by minimizing the Kullback–Leibler divergence between a target similarity distribution (e.g., personalized PageRank) and its low-dimensional approximation. Force2Vec (Rahman et al., 2020) reformulates the objective with linear-algebraic computations to enable high parallelism. Cui et al. (Cui et al., 2022) show that DeepWalk-based positional embeddings are strong initial artificial attributes for training GNNs, outperforming alternatives such as eigen, PageRank, or degree based attributes in several settings. RAHG (Li et al., 2023) constructs a role aware hypergraph with attention and residual connections, capturing both role information and adjacency while mitigating over-smoothing and modeling long range relations. However, random walk-based embeddings inherit intrinsic structural sampling bias, notably degree bias, which introduces distortions in synthesized attributes and degrades downstream GNN performance. To address this challenge, we propose FAKER. In contrast to prior work, FAKER uses PSD analysis to quantitatively diagnose sampling bias and adaptively re-parameterize the walk generation process itself. This yields a more balanced corpus and, in turn, higher-quality artificial attributes for GNN training.

## 3 METHODOLOGY

This paper proposes FAKER, a diagnosis-driven methodology for generating artificial attributes on non-attributed graphs. The workflow proceeds as follows. (i) FAKER converts random walks into group-wise visit signals by generating random walk sequences. (ii) These signals undergo analysis with PSD to diagnose exploration bias across groups. (iii) The sampler adapts its strategy by allocating additional walks to the biased groups according to the diagnosis. (iv) We train SGNS on the balanced walk corpus to obtain embeddings. Their combination with SI yields a fusion that forms the synthetic attribute matrix. Figure 3 summarizes the process.

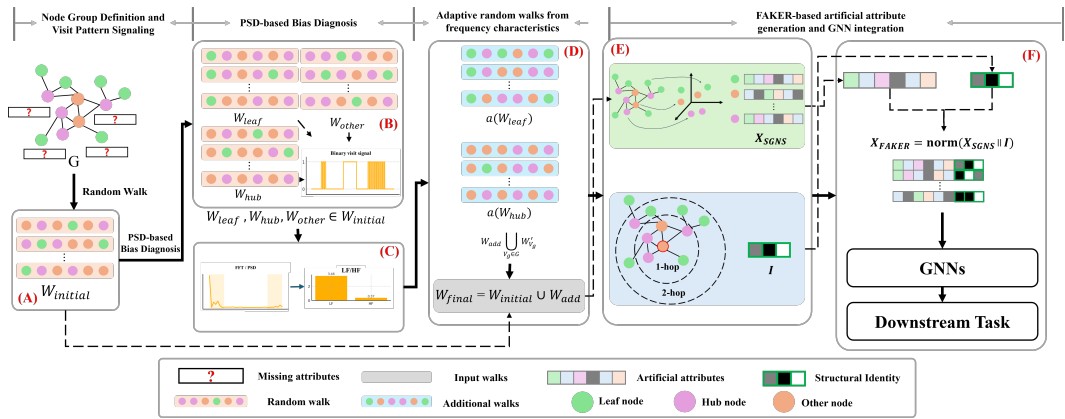

Figure 3: Overview of FAKER. (A) The framework groups nodes by structural role (hub/leaf/others). (B) It generates random walk sequences from each node, converting the sequences into per-group binary visit signals. (C) It applies group-wise PSD, producing LF/HF $z$-scores that diagnose persistence (LF) and switching (HF) biases. (D) These scores guide the allocation of additional walk sequences, merging the new sequences into the initial corpus. (E) The model trains SGNS on the final corpus to obtain $X_{\text{SGNS}}$. (F) Finally, it concatenates $X_{\text{SGNS}}$ with structural identity, yielding $X_{\text{FAKER}}$, which initializes the backbone GNN as $H^{(0)}$

## 3.1 PROBLEM DEFINITION

In non-attributed graphs, the node-feature matrix $X$ is absent. The task synthesizes an initial feature matrix from the adjacency $A$ for exclusive input to the backbone GNN. The formal definition expresses this process as

$$X_{\text{FAKER}} = h(A; \Psi) \in \mathbb{R}^{N \times d'}, \qquad H^{(0)} = X_{\text{FAKER}}.$$

where $h$ denotes the feature synthesizer (attribute generator). The training procedure fixes $h$ prior to downstream GNN training. A backbone GNN $f_\theta$ is then optimized for the task loss $\mathcal{L}_{\text{task}}(f_\theta(A, X_{\text{FAKER}}))$.

## 3.2 NODE GROUP DEFINITION AND VISIT PATTERN SIGNALING

FAKER analyzes random walk bias within node groups $V_g \subseteq V$, each containing structurally similar nodes. The model categorizes nodes into three structural roles consistent with prior work Kojaku et al. (2021); Liu et al. (2021). $V_{\text{leaf}}$ denotes the set of nodes with degree 1. These leaf nodes have low interconnectedness, resulting in fewer visits by the random walker. $V_{\text{hub}}$ denotes the set of nodes whose degree falls within the top 10% among all nodes. $V_{\text{other}}$ denotes the set of all remaining nodes that are not included in either $V_{\text{leaf}}$ or $V_{\text{hub}}$.

With every node as a starting point, the procedure generates $r$ random walks of length $L$, yielding $M = |V|r$ sequences $W_k = (w_0^{(k)}, \dots, w_{L-1}^{(k)})$.

For any group $V_g$, Eq. 1 encodes the group-level visit pattern of $W_k$ as a binary signal:

$$s_{k,g}(t) = \mathbf{1}\{ w_t^{(k)} \in V_g \}, \qquad t = 0, \dots, L-1. \tag{1}$$

The signal summarizes how frequently the walk visits $V_g$ and how long it persistence there versus switching across groups.

## 3.3 PSD-BASED BIAS DIAGNOSIS

Given the binary visit signals $s_{k,g}(t)$ from Sec. 3.2, the PSD of these signals quantifies group-specific exploration bias. For walk sequence $k$ and group $g$, let $\text{PSD}_{k,g}(f)$ denote the power spectral density of the mean-removed visit signal (i.e., with the zero-frequency/DC component removed).

Eq. 2 defines the representative spectrum for group $g$ through averaging over all walks.

$$\overline{\mathrm{PSD}}_g(f) = \frac{1}{M} \sum_{k=1}^{M} \mathrm{PSD}_{k,g}(f). \tag{2}$$

Let $\mathcal{F}_{\mathrm{LF}}$ and $\mathcal{F}_{\mathrm{HF}}$ denote the sets of the $n$ lowest non-DC and $n$ highest frequency bins, respectively (the DC bin at $f = 0$ is excluded). Eq. 3 denotes the corresponding average powers.

$$P_{\mathrm{LF}}(g) = \frac{1}{|\mathcal{F}_{\mathrm{LF}}|} \sum_{f \in \mathcal{F}_{\mathrm{LF}}} \overline{\mathrm{PSD}}_g(f), \qquad P_{\mathrm{HF}}(g) = \frac{1}{|\mathcal{F}_{\mathrm{HF}}|} \sum_{f \in \mathcal{F}_{\mathrm{HF}}} \overline{\mathrm{PSD}}_g(f). \tag{3}$$

For numerical stability, we then apply a log transform with a small constant $\varepsilon$:

$$L_g = \log\big(P_{\mathrm{LF}}(g) + \varepsilon\big), \qquad H_g = \log\big(P_{\mathrm{HF}}(g) + \varepsilon\big). \tag{4}$$

Finally, we standardize these log-power values across the set of groups $\mathcal{G}$ to obtain the final z-scores:

$$Z_{\mathrm{LF}}(g) = \frac{L_g - \mu_L}{\sigma_L}, \qquad Z_{\mathrm{HF}}(g) = \frac{H_g - \mu_H}{\sigma_H}, \tag{5}$$

where $\mu_L = \frac{1}{|\mathcal{G}|} \sum_{g \in \mathcal{G}} L_g$ and $\sigma_L$ is the standard deviation of $\{L_g\}_{g \in \mathcal{G}}$ (defined analogously for $H$). Large $Z_{\mathrm{LF}}(g)$ indicates prolonged persistence within group $g$, whereas large $Z_{\mathrm{HF}}(g)$ indicates frequent switching across groups. These statistics guide the adaptive walk strategy in Sec. 3.4.

### 3.4 ADAPTIVE RANDOM WALKS FROM FREQUENCY CHARACTERISTICS

The walk strategy for each group adapts the $z$-scores from Sec. 3.3. In Node2Vec, an increase in the return parameter $p$ discourages immediate backtracking. An increase in the in–out parameter $q$ biases the walk toward local BFS, while a decrease in $q$ promotes exploratory DFS.

**Parameter mapping.** Using a threshold $\tau > 0$ and a sensitivity parameter $\lambda_{\mathrm{pq}} > 0$, we define the piecewise mapping as follows:

$$\phi(z; \lambda_{\mathrm{pq}}, \tau) = \begin{cases} \frac{1}{|z| \lambda_{\mathrm{pq}}}, & z < -\tau, \\ 1, & |z| \leq \tau, \\ z \, \lambda_{\mathrm{pq}}, & z > \tau, \end{cases} \tag{6}$$

Eq. 7 denotes the node2vec parameters from the LF/HF scores.

$$p(g) = \mathrm{clip}\big(\phi(Z_{\mathrm{LF}}(g), \lambda_{\mathrm{pq}}, \tau), 0.1, 8\big), \qquad q(g) = \mathrm{clip}\big(\phi(Z_{\mathrm{HF}}(g), \lambda_{\mathrm{pq}}, \tau), 0.1, 8\big), \tag{7}$$

where $\mathrm{clip}(x, a, b) = \min\{\max\{x, a\}, b\}$. If $|Z_{\mathrm{LF}}(g)| \leq \tau$ then $p(g) = 1$ (no LF-based correction). If $|Z_{\mathrm{HF}}(g)| \leq \tau$ then $q(g) = 1$ (no HF-based correction). If both are within the threshold, it does not schedule any additional walks for $g$.

**Budget allocation.** Eq. 8 allocates the additional walks where bias is strongest:

$$a(g) = \Big\lfloor \lambda_{\mathrm{train}} \max\big\{ \mathbf{1}_{\{|Z_{\mathrm{LF}}(g)| > \tau\}} |Z_{\mathrm{LF}}(g)|, \ \mathbf{1}_{\{|Z_{\mathrm{HF}}(g)| > \tau\}} |Z_{\mathrm{HF}}(g)| \big\} \Big\rfloor, \tag{8}$$

The scalar $\lambda_{\mathrm{train}} > 0$ is a hyperparameter that controls the overall strength of the augmentation. When both scores exceed the threshold, it computes $a(g)$ with the larger magnitude.

**Adaptive sampling and final corpus.** For given input graph $G = (V, E)$, $\mathcal{N}(v) = \{x \in V : (v, x) \in E\}$ denotes the neighbor set of $v$. For each $g$ with $a(g) > 0$, our model generates $a(g)$ additional random walks $W'_g$. Each walk $W = (w_0, \ldots, w_{L-1})$ starts from a node drawn uniformly in the group, $w_0 \sim \mathcal{U}(V_g)$. For the first step ($i=1$), the transition is uniform over the neighbor set $\mathcal{N}(w_0) = \{x \in V : A_{w_0 x} = 1\}$. For steps $i \geq 2$, Eq. 9 use the second-order node2vec transition with group-specific $(p(g), q(g))$, where $u = w_{i-2}$ and $v = w_{i-1}$.

$$P\big(w_i = x \,\big|\, w_{i-1} = v, w_{i-2} = u\big) = \frac{\alpha_{p(g),q(g)}(u, x)}{\sum_{x' \in \mathcal{N}(v)} \alpha_{p(g),q(g)}(u, x')}, \qquad x \in \mathcal{N}(v), \tag{9}$$

where the search bias is

$$\alpha_{p,q}(u,x) = \begin{cases} 1/p, & d_{ux} = 0, \\ 1, & d_{ux} = 1, \\ 1/q, & d_{ux} = 2, \end{cases} \tag{10}$$

and $d_{ux} \in \{0, 1, 2\}$ denotes 0 if $x=u$, 1 if $(u,x) \in E$, and 2 otherwise. Eq. 11 merges the added walks with the initial corpus:

$$W_{\text{add}} = \bigcup_{g \in \mathcal{G}} W'_g, \qquad W_{\text{final}} = W_{\text{initial}} \cup W_{\text{add}}. \tag{11}$$

### 3.5 FAKER-BASED ARTIFICIAL ATTRIBUTE GENERATION AND GNN INTEGRATION

**Artificial attribute synthesis.** We train a standard skip-gram with negative sampling (SGNS) model on the balanced walk corpus $W_{\text{final}}$ to learn node embeddings. A symmetric window of size $T$ slides over each walk to form positive center–context pairs $\mathcal{D} = \{(i,j)\}$, where $i = w_t$ and $j \in \{t - T, \ldots, t-1, t+1, \ldots, t+T\}$. Given a noise distribution $p_0(\cdot)$ and $k$ negatives per positive, SGNS maximizes

$$\max_{\{u_i, v_j\}} \sum_{(i,j) \in \mathcal{D}} \Big[ \log \sigma(u_i^\top v_j) + \sum_{\ell=1}^{k} \log \sigma\big(-u_i^\top v_{n_\ell}\big) \Big], \qquad n_\ell \sim p_0, \tag{12}$$

where $u_i, v_j \in \mathbb{R}^{d'}$ denotes the center and context embeddings, and $\sigma(\cdot)$ denotes the sigmoid function. After training, the model outputs the synthetic attribute matrix with row-wise stacking of the center embeddings.

$$X_{\text{SGNS}} \in \mathbb{R}^{N \times d'}. $$

**Structural Identity.** We define the $k$-hop shell and its degree multiset as:

$$\mathcal{N}_k(v) = \{ u \in V : \text{dist}(u,v) = k \}, \qquad D_k(v) = \{ d(u) : u \in \mathcal{N}_k(v) \}. \tag{13}$$

For $k = 1, \ldots, K$, we summarize $D_k(v)$ by

$$\text{stats}(D_k(v)) = \big[ \min D_k(v), \ \max D_k(v), \ \text{mean} \, D_k(v), \ \text{std} \, D_k(v) \big], \tag{14}$$

and if $\mathcal{N}_k(v) = \varnothing$, we set $\text{stats}(D_k(v)) = [0,0,0,0]$. The SI vector is then

$$I(v) = \big[d(v)\big] \oplus \text{stats}\big(D_1(v)\big) \oplus \cdots \oplus \text{stats}\big(D_K(v)\big) \in \mathbb{R}^{1+4K}, \tag{15}$$

where $\oplus$ denotes concatenation, $I(v)$ denotes the structural identity vector that summarizes the local structural role of each node, and $d(v)$ denotes the degree of $v$. It yields $\mathbf{I} \in \mathbb{R}^{N \times (1+4K)}$ by stacking $I(v)$ over all nodes . Our experiments set $K=2$, so $\mathbf{I}$ has $1 + 4 \times 2 = 9$ columns.

**GNN integration.** The procedure constructs the final artificial attributes by concatenating the dynamic walk-based attributes with the static structural identity, followed by column-wise standardization:

$$X_{\text{FAKER}} = \text{norm}\big(X_{\text{SGNS}} \parallel \mathbf{I}\big), \qquad H^{(0)} = X_{\text{FAKER}}. \tag{16}$$

We freeze the generator before downstream training. The chosen backbone GNN then uses $X_{\text{FAKER}}$ as its input feature matrix.

## 4 EXPERIMENTS

### 4.1 EXPERIMENT SETTINGS

Our experimental evaluation is designed to answer the following research questions:

RQ1: On non-attributed graphs, does FAKER outperform structure-only baselines(A) on node classification and link prediction? How does it compare to X+A methods? RQ2: Which components of FAKER are responsible for the gains? (ablation) RQ3: How robust is FAKER to group definitions and hyperparameters?

Table 1: Node-classification accuracy on four benchmarks (%). Conventions: **A** denotes adjacency-only. **X+A** uses observed attributes. E denotes embedding-only. **Bold** marks the best, and underlined marks the second best. The symbol "—" indicates results unavailable because the original preprocessing is unreproducible.

| | method | Cora | Citeseer | Photo | Computer |
|---|---|---|---|---|---|
| X+A | SAT-GCN | 83.27 | 65.99 | 91.63 | 85.19 |
| | SAT-GAT | 85.79 | 67.67 | 92.6 | 87.66 |
| | WAGE | 85.9 | 69.33 | 92.4 | 88.67 |
| | Amer | 80.21 | 66.95 | 92.53 | 88.89 |
| | SVGA | 84.9 | 68.44 | 92.53 | 88.89 |
| | RITR | 85.81 | 69.01 | 92.24 | 88.49 |
| | AIAE-GCN | 85.34 | 69.15 | 92.18 | 87.78 |
| | AIAE-GAT | 85.75 | 69.46 | 92.06 | 86.68 |
| | MATE | 85.83 | 69.19 | 92.57 | 89.51 |
| | TDAR | 85.97 | 68.9 | 92.94 | 90.47 |
| | PCFI | 84.83 | **72.86** | 91.60 | 84.49 |
| | GCN(50%) | 78.11±6.83 | 68.48±5.74 | 90.53±2.10 | 87.49±2.41 |
| | GAT(50%) | 79.40±6.19 | 68.69±5.50 | 90.25±2.50 | 87.22±2.39 |
| | GraphSAGE(50%) | 73.31±8.98 | 62.96±7.72 | 85.40±5.55 | 79.82±5.27 |
| E | DeepWalk | 83.78 | 66.52 | 91.83 | 87.90 |
| | VERSE | 81.72 | 59.87 | 91.70 | 88.39 |
| | Force2Vec | 83.24 | 60.84 | — | — |
| | Residual2Vec | 81.20 | 56.75 | 92.71 | 90.00 |
| | FAKER-E | 84.91±0.27 | 72.42±0.64 | 93.04±0.28 | 90.19±0.06 |
| A | Cui-GCN | 85.93 | 70.34 | 92.97 | 90.53 |
| | Cui-GAT | 86.28 | 71.24 | 93.46 | 91.48 |
| | Cui-GraphSAGE | 86.04 | 69.56 | 92.46 | 90.80 |
| | RAHG | 85.82 | 72.24 | 93.33 | 87.68 |
| | FAKER-GCN | **87.02±0.33** | 72.69±0.27 | 93.41±0.12 | 91.86±0.05 |
| | FAKER-GAT | 86.87±0.42 | 72.56±0.16 | **93.52±0.09** | **91.91±0.02** |
| | FAKER-GraphSAGE | 86.77±0.22 | 72.53±0.50 | 92.96±0.12 | 90.71±0.11 |

**Datasets.** We evaluate on four benchmarks spanning two domains citation and recommendation (co-purchase) networks. See Appendix A.1 for dataset details.

**Baselines.** We compare our method against three categories of baselines: (i) non-attributed methods (A), including Cui et al. (2022) and RAHG (Li et al., 2023). (ii) node embedding models (E) Deep-Walk (Perozzi et al., 2014), VERSE (Tsitsulin et al., 2018), Force2Vec (Rahman et al., 2020) and Residual2Vec (Kojaku et al., 2021), and (iii) attribute-missing (X+A) methods, such as SAT (Chen et al., 2020), WAGE (Tu et al., 2025), AMER (Jin et al., 2022), SVGA (Yoo et al., 2022), RITR (Tu et al., 2024), AIAE (Xia et al., 2024), MATE (Peng et al., 2024), TDAR (Li et al., 2025), and PCFI (Um et al., 2023). The experiments evaluate FAKER on both node classification and link prediction tasks using three distinct backbone models: GCN (Kipf & Welling, 2016), GAT (Veličković et al., 2017), and GraphSAGE (Hamilton et al., 2017). Appendix A.2 offers an introduction to the compared models.

**Implementation Details.** Random walk based models (DeepWalk, Cui, RAHG) use the same walk-token budget as FAKER. Appendix A.4 describes the budget-matching protocol. The hyperparameters used in this paper are listed in Appendix A.3. Our experiments run on a single workstation with an NVIDIA GeForce RTX 3090 (24 GB) and an Intel Core i9-13900K (24 cores / 32 threads).

**Node classification.** Experiments use a transductive setup with 5-fold node-level cross-validation (80/20 per fold, Appendix A.5). Accuracy (mean±std across folds and 10 seeds) is reported. For X+A baselines, published numbers following each paper's protocol are cited. These are for context only and not directly comparable to the non-attributed (A) setting.

**Link Prediction.** To prevent information leakage, all walk generation, PSD diagnosis, and SGNS training are restricted to the training graph $G_{\text{train}} = (V, E_{\text{train}})$. We evaluate with AUC and AP, and report mean±std over 10 random seeds. Edge partitions follow prior work with a 60/20/20 split into train/validation/test sets (Chen et al., 2020). Appendix A.6 provides a detailed account of the safeguards.

Table 2: Link prediction on four benchmarks (AUC/AP). We compute all walks and embeddings on the training graph $G_{\text{train}}$, where validation and test edges are removed.

| | Method | Cora | | Citeseer | | Photo | | Computer | |
|---|---|---|---|---|---|---|---|---|---|
| | | AUC | AP | AUC | AP | AUC | AP | AUC | AP |
| X+A | SAT-GCN | 0.855 | 0.850 | 0.857 | 0.857 | 0.947 | 0.939 | 0.943 | 0.937 |
| | SAT-GAT | 0.893 | 0.902 | 0.892 | 0.914 | 0.928 | 0.911 | 0.910 | 0.894 |
| | Amer | 0.913 | 0.923 | 0.825 | 0.867 | 0.979 | 0.978 | 0.964 | 0.961 |
| | PCFI | 0.822 | 0.852 | 0.800 | 0.832 | 0.716 | 0.649 | 0.562 | 0.538 |
| | GCN(50%) | $0.677_{\pm0.011}$ | $0.702_{\pm0.014}$ | $0.786_{\pm0.058}$ | $0.793_{\pm0.068}$ | $0.936_{\pm0.057}$ | $0.931_{\pm0.058}$ | $0.799_{\pm0.017}$ | $0.816_{\pm0.013}$ |
| | GAT(50%) | $0.697_{\pm0.041}$ | $0.680_{\pm0.024}$ | $0.812_{\pm0.001}$ | $0.805_{\pm0.003}$ | $0.916_{\pm0.088}$ | $0.915_{\pm0.086}$ | $0.985_{\pm0.001}$ | $0.983_{\pm0.001}$ |
| | GraphSAGE(50%) | $0.893_{\pm0.006}$ | $0.920_{\pm0.005}$ | $0.913_{\pm0.002}$ | $0.939_{\pm0.001}$ | $0.973_{\pm0.002}$ | $0.966_{\pm0.002}$ | $0.973_{\pm0.002}$ | $0.966_{\pm0.002}$ |
| E | DeepWalk | 0.728 | 0.803 | 0.637 | 0.744 | 0.965 | 0.957 | 0.940 | 0.935 |
| | Residual2Vec | 0.567 | 0.588 | 0.566 | 0.552 | 0.516 | 0.575 | 0.551 | 0.586 |
| | FAKER-E | $0.928_{\pm0.004}$ | $0.950_{\pm0.002}$ | $0.919_{\pm0.001}$ | $0.944_{\pm0.001}$ | $0.981_{\pm0.001}$ | $0.977_{\pm0.001}$ | $0.976_{\pm0.001}$ | $0.974_{\pm0.001}$ |
| A | Cui-GCN | $0.721_{\pm0.004}$ | $0.798_{\pm0.002}$ | $0.717_{\pm0.019}$ | $0.776_{\pm0.011}$ | $0.982_{\pm0.002}$ | $0.981_{\pm0.002}$ | $0.978_{\pm0.001}$ | $0.978_{\pm0.001}$ |
| | Cui-GAT | $0.767_{\pm0.014}$ | $0.789_{\pm0.013}$ | $0.719_{\pm0.004}$ | $0.736_{\pm0.037}$ | $0.976_{\pm0.003}$ | $0.972_{\pm0.004}$ | $0.968_{\pm0.003}$ | $0.964_{\pm0.004}$ |
| | Cui-GraphSAGE | $0.742_{\pm0.020}$ | $0.789_{\pm0.013}$ | $0.682_{\pm0.022}$ | $0.720_{\pm0.024}$ | $0.981_{\pm0.001}$ | $0.979_{\pm0.001}$ | $0.979_{\pm0.001}$ | $0.979_{\pm0.001}$ |
| | RAHG | 0.789 | 0.804 | 0.764 | 0.796 | 0.960 | 0.951 | 0.947 | 0.938 |
| | FAKER-GCN | $\mathbf{0.943}_{\pm0.002}$ | $\mathbf{0.958}_{\pm0.001}$ | $0.935_{\pm0.002}$ | $0.953_{\pm0.001}$ | $0.993_{\pm0.001}$ | $0.992_{\pm0.001}$ | $\mathbf{0.994}_{\pm0.001}$ | $\mathbf{0.994}_{\pm0.001}$ |
| | FAKER-GAT | $0.939_{\pm0.001}$ | $\mathbf{0.958}_{\pm0.001}$ | $0.925_{\pm0.002}$ | $0.949_{\pm0.002}$ | $\mathbf{0.995}_{\pm0.002}$ | $\mathbf{0.994}_{\pm0.001}$ | $0.993_{\pm0.001}$ | $0.992_{\pm0.001}$ |
| | FAKER-GraphSAGE | $0.924_{\pm0.021}$ | $0.945_{\pm0.019}$ | $\mathbf{0.949}_{\pm0.003}$ | $\mathbf{0.960}_{\pm0.002}$ | $0.992_{\pm0.001}$ | $0.990_{\pm0.001}$ | $0.993_{\pm0.001}$ | $0.992_{\pm0.001}$ |

## 4.2 PERFORMANCE COMPARISON (RQ1)

**Node classification (Table 1).** FAKER achieves the best non-attributed (A) accuracy on all four datasets, improving over the strongest **A** baselines by +0.74 (Cora), +0.45 (Citeseer), +0.06 (Photo), and +0.43 (Computer). The embedding-only variant FAKER-E also outperforms node-embedding baselines with gains of +1.13, +5.90, +1.21, and +2.29, respectively. Compared with 50% feature-masked GNNs, the best FAKER variant improves Accuracy by +7.62–13.71 on Cora, +4.00–9.73 on Citeseer, +2.99–8.12 on Photo, and +4.42–12.09 on Computer. Notably, GNN (50%) baselines that randomly mask nodes show large performance variation depending on which nodes remain. In contrast, FAKER is highly stable across random seeds. This consistency indicates that FAKER leverages structural information to synthesize reliable, high-quality artificial attributes for all nodes. Furthermore, FAKER not only competes with but also outperforms the majority of X+A methods that have access to partial attributes. Appendix A.5 details the performance comparison between FAKER and GNN models under varying attribute missing rates.

**Link prediction (Table 2).** FAKER achieves the best AUC/AP among adjacency-only (A) methods across four datasets. Performance on the co-purchase graphs (Photo, Computer) is nearly perfect (AUC/AP≈0.99). On the citation graphs, the gains over the strongest A-only baseline are substantial(+0.154 AUC / +0.154 AP on Cora and +0.185 / +0.164 on Citeseer). On the co-purchase graphs, the margins are smaller but consistent(+0.013 / +0.013 on Photo and +0.015 / +0.015 on Computer). Furthermore, the embedding-only variant FAKER-E achieves strong performance independently. The learning signal for low-degree nodes and inter-group connections improves because PSD-based correction reduces hub over-sampling and low-degree under-sampling. These results indicate that PSD-guided walk control and SI-augmented features enable strong link recovery without access to raw attributes.

## 4.3 ANALYSIS OF FAKER'S COMPONENTS (RQ2)

Table 3 analyzes the performance contribution of FAKER's two core components, PSD-based diagnosis/correction and SI fusion. Removing the PSD-based bias correction produces the largest and most consistent drop across all datasets. The declines substantially exceed each setting's standard deviation, indicating an effect well beyond stochastic variation. In addition, PSD correction improves not only the mean performance but also reduces the standard deviation itself, thereby enhancing stability. SI fusion yields smaller yet uniform gains on every dataset. This suggests that SI effectively complements the SGNS embeddings by supplying local, static structural information that the walk-trained features alone struggle to capture. Detailed ablations for link prediction are provided in Appendix A.7.

Table 3: Ablation study on the effectiveness of each component of FAKER. "FAKER w/o PSD" denotes SGNS embeddings from standard random walks without PSD correction. "FAKER w/o SI" denotes bias-corrected SGNS embeddings without SI fusion.

| Method | Cora | Citeseer | Photo | Computer |
|---|---|---|---|---|
| FAKER-GCN w/o PSD | 85.49±0.55 | 69.73±0.71 | 92.29±0.25 | 90.20±0.09 |
| FAKER-GCN w/o SI | 86.83±0.36 | 72.41±0.45 | 93.18±0.25 | 91.53±0.07 |
| FAKER-GCN | **87.02±0.33** | **72.69±0.27** | **93.41±0.12** | **91.86±0.05** |

## 4.4 ROBUSTNESS TO GROUP DEFINITIONS AND HYPERPARAMETERS (RQ3)

This section assesses the sensitivity of FAKER to how node groups are defined and to key training/diagnosis hyperparameters.

**Group-definition variants.** We vary (i) hub thresholds $V_{\text{hub}} \in \{1\%, 3\%, 5\%, 10\%, 20\%\}$, (ii) leaf rules with degree cutoffs $V_{\text{leaf}} \in \{2, 3, 5\}$ and ranges [1-2], [1-3], [1-5], and (iii) equal-size partitions by betweenness/eigen vector/PageRank.

**Findings-group definitions.** Expanding the hub fraction to 20% produces the largest performance drop. An overly broad hub set blurs role boundaries across groups, which blocks effective correction of the high-degree exploration bias we target. In contrast, changing the leaf rule (cutoffs or ranges) has only minor effects. Even after widening the leaf range (increases walk tokens) accuracy decreases slightly. This pattern suggests that the visit-signal/PSD diagnosis already captures low-degree regions well, and that naive range expansion adds noise rather than useful signal. Across these group-definition scenarios, FAKER consistently and clearly outperforms the Cui-GCN baseline. The gains thus stem from the core PSD-guided bias-correction mechanism, not from a hand-picked group definition. Appendix B reports additional sensitivity analyses and robustness checks over hyperparameters.

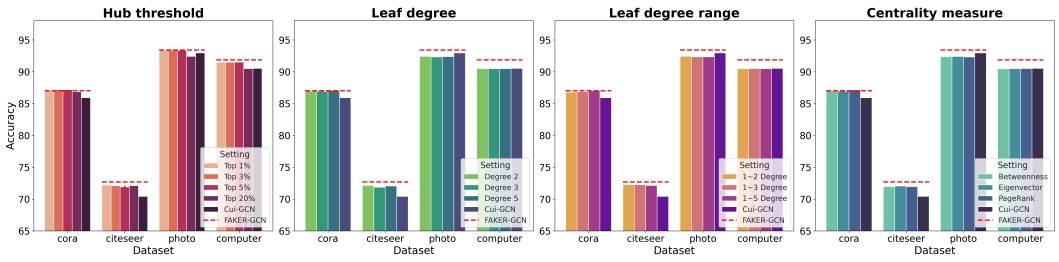

Figure 4: Robustness to group definitions. The red dashed line marks the default FAKER–GCN.

## 5 CONCLUSION

This paper introduces FAKER, a diagnosis-driven framework that synthesizes balanced artificial attributes for non-attributed graphs. By analyzing group-level visit signals in the PSD, FAKER quantifies persistence and switching biases. It adapts the random-walk strategy accordingly to build a balanced corpus. The resulting features integrate with standard GNNs without architectural changes. Across four benchmarks and under matched walk-token budgets, FAKER consistently leads adjacency-only baselines on node classification and link prediction. It also rivals feature-using methods in many cases. Ablation and robustness studies trace the gains to frequency-guided allocation rather than walk volume or delicate hyperparameter tuning. These results position FAKER as a simple and reliable tool for feature-scarce settings.

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

# A EXPERIMENTS

## A.1 DATASETS

The datasets used in our experiments are standard benchmarks provided by the PyTorch Geometric package. Statistics are summarized in Table 4.

Table 4: Statistics of the benchmark datasets, where "Avg Hot Num" denotes the average number of active entries in the multi-hot node attributes.

| Dataset | Nodes | Edges | Attribute Dim | Avg Hot Num | Classes |
|---------|-------|-------|---------------|-------------|---------|
| **Cora** | 2,708 | 5,278 | 1,433 | 18.17 | 7 |
| **Citeseer** | 3,327 | 4,228 | 3,703 | 31.60 | 6 |
| **Photo** | 7,650 | 119,081 | 745 | 258.81 | 8 |
| **Computer** | 13,752 | 245,861 | 767 | 267.23 | 10 |

- **Cora, Citeseer.** These are citation networks where nodes represent academic publications and edges represent citations. Node features are sparse, binary bag-of-words vectors indicating the presence or absence of keywords. The node labels represent the publication's research area.

- **Amazon Computers, Photo.** These are co-purchase networks where nodes represent products. An edge between two nodes indicates that they are frequently bought together.The node labels are derived from product categories.

## A.2 INTRODUCTION OF BASELINES

**Attribute-completion (X+A).**

- **SAT Chen et al. (2020)**: This is a distribution-matching framework that aligns structure-derived and attribute-derived embeddings in a shared latent space to impute missing node features while remaining plug-and-play with common GNN backbones.
- **WAGE Tu et al. (2025)**: This is a weight-distribution encoder that tightly couples topology and attributes to reconstruct node features reliably under high missingness.
- **AMER Jin et al. (2022)**: This is a joint learning scheme that completes attributes and learns representations together, using mutual-information maximization with an adversarial consistency objective.
- **SVGA Yoo et al. (2022)**: This is a structured variational approach that employs a Gaussian Markov random field prior to model inter-feature dependencies and complete missing attributes.
- **RITR Tu et al. (2024)**: This is an initialize-then-refine pipeline tailored for mixed missingness, applying distinct strategies to attribute-missing and attribute-incomplete cases.
- **AIAE Xia et al. (2024)**: This is a dual-encoder design with knowledge distillation that fuses structural and attribute cues to denoise inputs and enhance imputation expressiveness.
- **MATE Peng et al. (2024)**: This is a dual-consistency method that jointly optimizes input-space attributes and latent codes by enforcing view-wise (structure/attribute) agreement.
- **TDAR Li et al. (2025)**: This is a topology-guided denoising and attribute reconstruction framework that regularizes both structure and features to yield robust completions.
- **PCFI Um et al. (2023)**: This is a pseudo-confidence–driven imputer that assigns channel-wise reliabilities so that confident feature channels refine uncertain ones.

**Node-embedding (E).**

- **DeepWalk Perozzi et al. (2014)**: This is a random-walk–based method that trains skip-gram on node sequences to produce unsupervised node embeddings widely used as synthetic attributes.
- **VERSE Tsitsulin et al. (2018)**: This is a similarity-preserving embedder that minimizes KL divergence between a chosen target similarity (e.g., PPR) and its low-dimensional approximation.
- **Residual2Vec Kojaku et al. (2021)**: This is a debiasing framework that models and subtracts random-walk biases via a null-graph baseline, yielding residual (degree/structure-agnostic) embeddings that improve link prediction and clustering.

**Adjacency-only (A).**

- **RAHG Li et al. (2023)**: This is a role-aware hypergraph encoder with attention and residual connections that captures both proximity and structural roles while mitigating over-smoothing and enabling long-range interactions.
- **Cui Cui et al. (2022)**: This is a practice-oriented pipeline showing that random-walk embeddings used directly as initial node features provide strong, general-purpose inputs for downstream GNNs.

**Backbone GNNs.**

- **GCN Kipf & Welling (2016)**: This is a seminal architecture that integrates node attributes with graph structure through localized, diffusion-style spectral convolutions.

- **GAT Veličković et al. (2017)**: This is an attention-based GNN that learns edge-specific importance weights to adaptively aggregate neighborhood information.
- **GraphSAGE Hamilton et al. (2017)**: This is a scalable neighborhood-sampling framework that aggregates sampled neighbors with learnable functions to update node representations.

### A.3 HYPERPARAMTERS

Table 5 lists the hyperparameter settings for our experiments.

Table 5: Hyperparameter settings.

| Hyperparameter | Symbol (in paper) | Value |
|---|---|---|
| *GNN Training Parameters* | | |
| GNN Layers | | 2 |
| Weight Decay | | $5 \times 10^{-4}$ |
| Learning Rate | | 0.01 |
| Learning Epochs | | 1000 |
| Early Stopping Patience | | 25 |
| Number of GNN Layers | | 2 |
| GNN Hidden Size | | 256 |
| Optimizer | | Adam |
| Dropout Rate | | 0.5 |
| GAT Attention Head | | 4 |
| Activation Function | | ReLU |
| *FAKER-specific Parameters* | | |
| Random Walk Epochs | $r$ | 40 |
| Walk Length | $L$ | 20 |
| SGNS Embedding Dimension | $d'$ | 256 |
| SGNS Training Epochs | | 10 |
| SGNS Window Size | $T$ | 10 |
| SGNS Negative Samples | $k$ | 5 |
| Negative Sampling Exponent | | 0.75 |
| SGNS Batch Size (tokens) | | 128 |
| Structural Identity Max Hops | $K$ | 2 |
| PSD Frequency Bins | $n$ | 3 |
| (p,q) Sensitivity | $\lambda_{pq}$ | 5 |
| Additional Walk Weight | $\lambda_{\text{train}}$ | 8 |
| Frequency Threshold | $\tau$ | 0.7 |

### A.4 RANDOM-WALK BUDGET PARITY

For a fair comparison, FAKER and all random walk based baselines (e.g., DeepWalk, Residual2Vec, RAHG) use the same number of walks per node on each dataset. We always generate complete walks of fixed length $L$ without truncation. The budget depends on walks per node rather than tokens; with $L$ fixed across methods, matching walks naturally matches tokens. We apply this procedure to the final corpus after the augmentation-and-merge in Eq. 11.

**Notation.** $N = |V|$ denotes the number of nodes, $L$ denotes the walk length, and $r_0$ denotes the initial walks-per-node. We denote the number of additional walks by $\Delta W$ after FAKER generates additional walks, merging them as in Eq. 11. Then final number of walks of FAKER is

$$|W_{\text{FAKER}}| = Nr_0 + \Delta W.$$

**Quotient–Remainder Decomposition.** We decompose the additional walks by dividing $\Delta W$ by $N$:

$$\Delta W = qN + \rho, \qquad q = \left\lfloor \frac{\Delta W}{N} \right\rfloor, \ 0 \leq \rho < N.$$

**Baseline Augmentation Protocol.** Suppose each baseline's initial corpus contains $Nr_0$ length-$L$ walks. We add exactly $\Delta W$ complete walks in two steps: Each baseline starts with an initial corpus of $Nr_0$ walks of length $L$. We then augment the corpus by adding exactly $\Delta W$ complete walks in two steps:

1. **Full-node passes.** We perform $q$ additional full passes, where each pass generates one length-$L$ walk per node, yielding additional $qN$ walks in total.

2. **Residual pass.** We then sample $\rho$ nodes from $V$ uniformly without replacement, with one length-$L$ walk generated from each to produce $\rho$ additional walks.

The final number of walks in baselines then becomes

$$|W_{\text{baseline}}| \;=\; Nr_0 + qN + \rho \;=\; Nr_0 + \Delta W \;=\; |W_{\text{FAKER}}|,$$

which is exactly equal to that of FAKER.

A.5 NODE CLASSIFICATION

The evaluation uses five-fold node-level cross-validation with an 80%/20% train/test split per fold. All methods are implemented in a unified codebase and executed under the same protocol to ensure a fair comparison. Each configuration is run with 10 random seeds, and results are reported as the mean. For the attribute-missing (X+A) setting, we follow the protocol from (Chen et al., 2020). In this protocol, nodes with observed attributes are split into 40% for training, 10% for validation, and 50% for testing. Baseline results for this setting are cited from their original papers.

While GNN models such as GCN, GAT, and GraphSAGE suffer from a sharp performance drop as attribute features are missing, FAKER maintains performance comparable to GNNs that utilize all attributes, despite not using any attribute information(Figure 5 Cora, Photo, Computer). This advantage manifests strongly for models sensitive to attribute quality, such as GraphSAGE. The performance of GraphSAGE collapses as features degrade, whereas FAKER performance remains high. These results show that the artificial attributes generated by FAKER possess quality sufficient to rival fully observed raw features. Consequently, FAKER is not only effective in non-attributed settings but also a reliable alternative in typical attribute-missing scenarios where observed features are sparse or noisy.

A.6 LINK PREDICTION PROTOCOL AND LEAKAGE SAFEGUARDS

We follow the 60/20/20 edge split with fixed seeds: $E = E_{\text{train}} \cup E_{\text{val}} \cup E_{\text{test}}$ and $G_{\text{train}} = (V, E_{\text{train}})$.

- Training $E_{\text{train}}$: used to build the message-passing graph and to train models.
- Validation $E_{\text{val}}$: used only for hyperparameter selection and early stopping; never added to the training graph.
- Test $E_{\text{test}}$: used only for final evaluation.

We also sample disjoint negative edges (non-links) for each split, with $|N_S| = |E_S|$ for $S \in \{\text{train, val, test}\}$. This strict separation guarantees that links evaluated at validation or test time are never observed during feature construction or training.

A.7 ABLATION VARIANT DEFINITIONS

**Ablation study on node classification.** In Table 6, the PSD-based bias correction exerts the most decisive influence on FAKER's performance. FAKER w/o PSD exhibits the largest and most consistent performance drop across all datasets. FAKER w/o SI shows a consistent yet relatively small decline across all datasets. This result demonstrates that SI complements the structural information lost during bias correction.

**Ablation study on link prediction.** In Table 7, the importance of PSD-based bias correction becomes more pronounced in link prediction. With a GCN backbone, w/o PSD shows an AUC drop of $-0.177$ on Cora and $-0.205$ on Citeseer. Results on Photo and Computer suggest that PSD-based correction substantially improves overall performance, although the additional gain can be

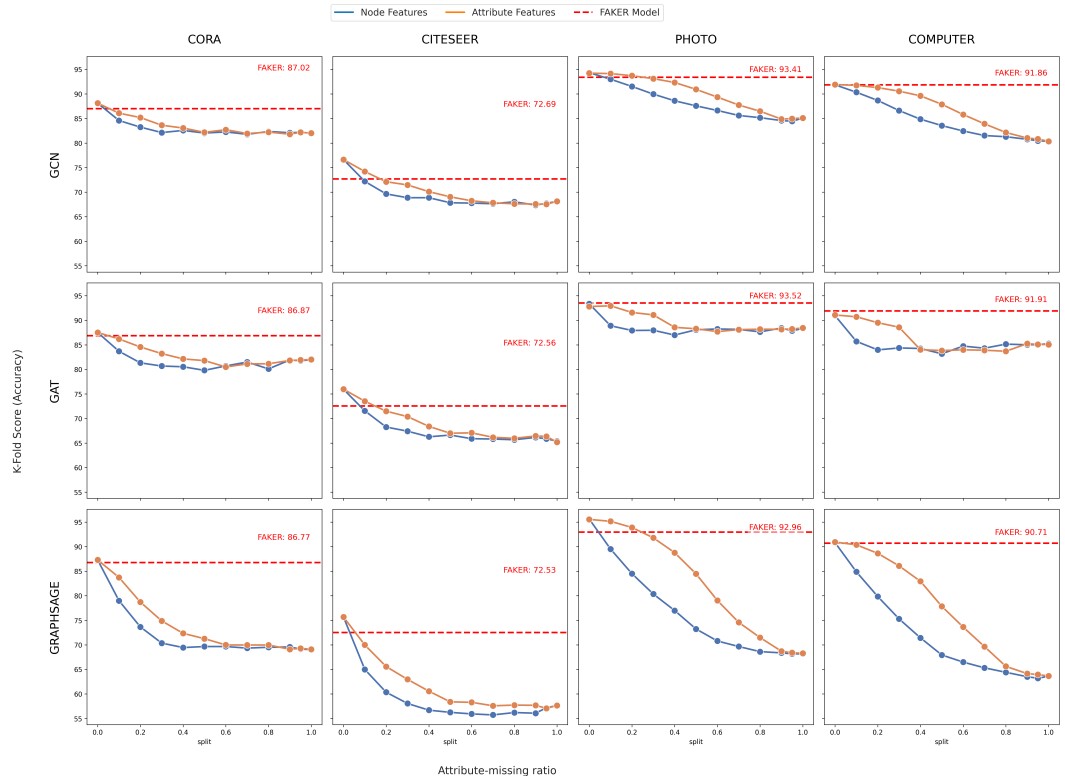

Figure 5: Accuracy comparison between GNN models and FAKER under different attribute-missing rates, where "-Node" replaces a ratio of nodes' feature vectors with noise and "-Attribute" replaces a ratio of feature dimensions with noise.

limited when the backbone model approaches saturation. The presence or absence of SI exerts only a marginal effect on link prediction. This finding indicates that in a topology-focused task such as link prediction, bias-corrected dynamic walk embeddings already capture most of the necessary information.

Table 6: Ablation study on node classification (Accuracy).

| Method | Cora | Citeseer | Photo | Computer |
|---|---|---|---|---|
| FAKER-GAT w/o PSD | 85.49±0.55 | 70.52±0.88 | 92.53±0.04 | 91.62±0.17 |
| FAKER-GAT w/o SI | 86.76±0.35 | 72.37±0.39 | 93.34±0.06 | 91.11±0.11 |
| FAKER-GAT | **86.87±0.42** | **72.56±0.16** | **93.52±0.09** | **91.91±0.02** |
| FAKER-GraphSAGE w/o PSD | 84.81±0.40 | 68.05±1.33 | 91.90±0.27 | 89.98±0.23 |
| FAKER-GraphSAGE w/o SI | 86.37±0.21 | 72.02±0.59 | 92.66±0.07 | 90.02±0.16 |
| FAKER-GraphSAGE | **86.77±0.22** | **72.53±0.50** | **92.96±0.12** | **90.71±0.11** |

## B HYPERPARAMETER SWEEPS

**Setup.** We fix all hyperparameters to the defaults in Appendix A.3 and vary one hyperparameter at a time: SGNS epochs (learning_epoch), SGNS embedding size (n2v_d), GNN hidden size (gnn_d), GNN epochs (epoch), $(p, q)$ sensitivity $\lambda_{pq}$ (pq), allocation weight $\lambda_{train}$ (train_weight), the threshold $\tau$ (freq_val), and walk length $L$ (walklen). All runs use FAKER–GCN under a fixed walk–token budget, and we report mean±std over 10 seeds.

Table 7: Ablation study on link prediction (AUC/AP).

| Method | Cora | | Citeseer | | Photo | | Computer | |
|---|---|---|---|---|---|---|---|---|
| | AUC | AP | AUC | AP | AUC | AP | AUC | AP |
| FAKER-GCN w/o PSD | $0.766_{\pm 0.017}$ | $0.791_{\pm 0.021}$ | $0.730_{\pm 0.014}$ | $0.778_{\pm 0.011}$ | $0.981_{\pm 0.001}$ | $0.979_{\pm 0.001}$ | $0.977_{\pm 0.001}$ | $0.977_{\pm 0.001}$ |
| FAKER-GCN w/o SI | $0.939_{\pm 0.002}$ | $\mathbf{0.958}_{\pm 0.001}$ | $0.917_{\pm 0.011}$ | $0.933_{\pm 0.014}$ | $0.987_{\pm 0.001}$ | $0.985_{\pm 0.001}$ | $0.993_{\pm 0.001}$ | $0.988_{\pm 0.001}$ |
| FAKER-GCN | $\mathbf{0.943}_{\pm 0.001}$ | $\mathbf{0.958}_{\pm 0.001}$ | $\mathbf{0.935}_{\pm 0.002}$ | $\mathbf{0.953}_{\pm 0.001}$ | $\mathbf{0.993}_{\pm 0.001}$ | $\mathbf{0.992}_{\pm 0.001}$ | $\mathbf{0.994}_{\pm 0.001}$ | $\mathbf{0.994}_{\pm 0.001}$ |
| FAKER-GAT w/o PSD | $0.763_{\pm 0.016}$ | $0.799_{\pm 0.016}$ | $0.722_{\pm 0.016}$ | $0.774_{\pm 0.011}$ | $0.976_{\pm 0.002}$ | $0.972_{\pm 0.003}$ | $0.976_{\pm 0.001}$ | $0.979_{\pm 0.001}$ |
| FAKER-GAT w/o SI | $0.938_{\pm 0.001}$ | $\mathbf{0.958}_{\pm 0.001}$ | $0.917_{\pm 0.001}$ | $0.944_{\pm 0.001}$ | $0.990_{\pm 0.001}$ | $0.985_{\pm 0.001}$ | $0.990_{\pm 0.001}$ | $0.988_{\pm 0.001}$ |
| FAKER-GAT | $\mathbf{0.939}_{\pm 0.001}$ | $\mathbf{0.958}_{\pm 0.001}$ | $\mathbf{0.925}_{\pm 0.002}$ | $\mathbf{0.949}_{\pm 0.002}$ | $\mathbf{0.995}_{\pm 0.002}$ | $\mathbf{0.994}_{\pm 0.001}$ | $\mathbf{0.993}_{\pm 0.001}$ | $\mathbf{0.992}_{\pm 0.001}$ |
| FAKER-GraphSAGE w/o PSD | $0.719_{\pm 0.096}$ | $0.738_{\pm 0.111}$ | $0.704_{\pm 0.021}$ | $0.747_{\pm 0.027}$ | $0.982_{\pm 0.003}$ | $0.982_{\pm 0.003}$ | $0.980_{\pm 0.001}$ | $0.979_{\pm 0.001}$ |
| FAKER-GraphSAGE w/o SI | $0.928_{\pm 0.001}$ | $\mathbf{0.945}_{\pm 0.001}$ | $0.917_{\pm 0.001}$ | $0.944_{\pm 0.001}$ | $0.991_{\pm 0.001}$ | $0.989_{\pm 0.001}$ | $\mathbf{0.993}_{\pm 0.001}$ | $\mathbf{0.992}_{\pm 0.001}$ |
| FAKER-GraphSAGE | $\mathbf{0.924}_{\pm 0.021}$ | $\mathbf{0.945}_{\pm 0.019}$ | $\mathbf{0.949}_{\pm 0.003}$ | $\mathbf{0.960}_{\pm 0.002}$ | $\mathbf{0.992}_{\pm 0.001}$ | $\mathbf{0.990}_{\pm 0.001}$ | $\mathbf{0.993}_{\pm 0.001}$ | $\mathbf{0.992}_{\pm 0.001}$ |

**Result.** (1) **Model dimension.** Increasing n2v_d and gnn_d yields small, monotonic gains up to typical sizes, then plateaus. (2) **Diagnosis & control.** Within broad ranges, $\lambda_{\mathrm{pq}}$ and $\tau$ have mild effects. mid-range settings work best across datasets. Using small values for the allocation weight (train_weight) resulted in performance degradation on certain datasets. (3) **Walk length.** Larger $L$ improves accuracy up to $L=40$. beyond that, returns diminish under a fixed token budget. (4) **Overall robustness.** Curves change smoothly and exhibit wide plateaus, indicating a low tuning burden. Unlike pipelines that require fragile, fine-grained hyperparameter search, FAKER remains stable across SGNS/GNN training knobs and diagnosis/actuation parameters.

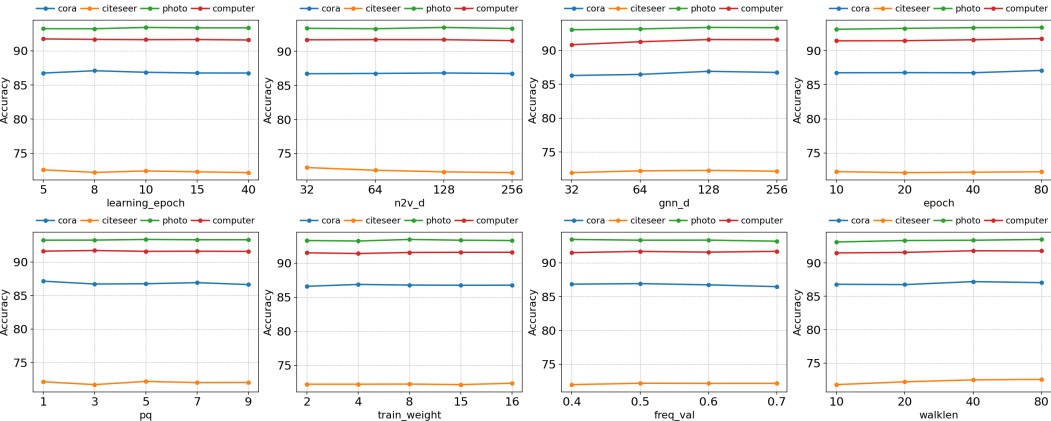

Figure 6: Robustness of our model under variations of training and bias-diagnosis hyperparameters.

