# OpenReview forum: "FAKER: Generating Frequency-based Artificial Attributes via Random Walks for Non-attribute Graph Representation Learning"
_ICLR.cc/2026/Conference — ICLR 2026 Conference Withdrawn Submission_

### Official Review · Reviewer_Uvtf · 2025-10-26

**Soundness:** 2
**Presentation:** 3
**Contribution:** 2
**Rating:** 2
**Confidence:** 4

**Summary:**

The paper proposes a node attribute generation framework designed primarily for Graph Neural Networks (GNNs). The paper relies on a random walk–based embedding mechanism and introduces a novel strategy to augment the set of walks by generating additional random walks in order to manipulate node occurrence frequencies. The generated random walks (i.e., node sequences) are then used to learn node embeddings using the SkipGram approach, which serves as node features for the GNN models. The approach is evaluated on node classification and link prediction tasks and outperforms baseline approaches across multiple datasets.

**Strengths:**

- The paper is generally well-organized and includes illustrative figures that help in understanding of the proposed framework.
- Addressing the problem of node featureless graphs is a meaningful and relevant challenge for the GNN community.
- The two-step random walk design offers an intuitive way to control node appearance frequency and its influence on embedding quality.

**Weaknesses:**

- The proposed framework largely builds upon existing random walk–based embedding approaches (e.g., DeepWalk, Node2Vec) and provides a limited methodological novelty.
- The paper lacks a deeper theoretical justification for the proposed components. Since the paper proposes a node attribute generation approach based on random walks, a GNN or an inductive method that uses these artificial features cannot handle new nodes inductively without re-training, undermining one of the key advantages of many GNN approaches.
- The datasets used (e.g., Cora, Citeseer) are quite small and sparse, and the train/validation/test splits for link prediction seem harsh. Some baseline results also lack standard deviation values.

**Questions:**

- In Table 1, FAKER-E performs similarly to the variants of the proposed approach except on the Cora dataset. Can the authors elaborate on whether the generated node features truly enhance GNN learning or mainly replicate the effects of existing embeddings?

- For the baseline models lacking standard deviations, were the results copied directly from the original papers? If that's the case, how comparable are the experimental setups?

- Could the authors clarify the motivation behind using a two-step random walk procedure? Why not apply a biased walk directly, since the node visiting frequency is degree-proportional for an unbiased walking strategy?

- Have the authors considered an alternative to explicit walk generation, such as reweighting node pairs during optimization to control co-occurrence frequencies?

- Could the authors explain the almost perfect link prediction scores on the Computer and Photo datasets? Are these results expected given dataset density or model bias?

- It is unclear why Node2Vec, a closely related random walk–based embedding method, was not included as a baseline.

- Are there any empty neighbor set cases as indicated in Eq. 302? Do the networks contain such isolated nodes or very small disconnected components?

- Several hyperparameters (e.g., a=0.1, b=8 in Eq. 7) and architectural decisions (e.g., maximum hop K=2) are not motivated or analyzed. An additional ablation study for these choices could be provided.

**Minor clarity issues:**
- The symbol, $\psi$ in Subsection 3.1 is undefined.
- Table and figure captions are sometimes incomplete (e.g., Table 3 does not specify that it refers to node classification results. Figure 1's caption lacks task indication).
- It is unclear which random walk generation method is used initially (uniform or biased).

---

### Official Review · Reviewer_6AKV · 2025-10-30

**Soundness:** 3
**Presentation:** 3
**Contribution:** 3
**Rating:** 6
**Confidence:** 3

**Summary:**

This paper tackles the problem of learning graph representations without initial node features, proposing FAKER, a model-agnostic framework that synthesizes artificial attributes from topology alone. The method diagnoses frequency-domain biases in random walks using power spectral density, then employs adaptive sampling and a lightweight co-occurrence encoder to generate node features for GNNs. The paper clearly defines the problem and presents a systematic solution combining bias diagnosis and attribute synthesis. Experimental results show consistent improvements over baselines and competitive performance with feature-based methods.

**Strengths:**

1. The issue studied in this article is novel.

2. The motivation of the article is clear.

3. The techniques used are solid.

**Weaknesses:**

1. The computational complexity of the method needs to be discussed, along with validation on large datasets.

2. Although scenarios with completely missing attributes exist, they are relatively rare. Comparisons with different baseline methods under varying missing rates could better validate the effectiveness of the method.

**Questions:**

Please see the weaknesses.

---

### Official Review · Reviewer_AuTj · 2025-11-01

**Soundness:** 2
**Presentation:** 3
**Contribution:** 2
**Rating:** 4
**Confidence:** 5

**Summary:**

This paper focuses the problem of creating base embeddings that can be used as node attributes for graph neural networks.  The authors identify that existing methods based on random walks suffer from inherent sampling biases (e.g., degree bias, where hubs are over-sampled and low-degree nodes are under-sampled). To tackle this problem, the authors propose FAKER that arguments the random walk data for training by balancing the walks with strong low-frequency and high-frequency components. The results are improved performance for node classification and link prediction tasks, with various baselines.

**Strengths:**

1. The framework is well-motivated. Instead of attempting to debias embeddings in a post-hoc manner (e.g., via the loss function), FAKER tackles the sampling bias "at its source" by adaptively correcting the walk generation process itself.
3. The experiments compared various baseline models across four datasets and confirmed the consistency of the results.
4. Conducted ablation study to identify the source of improvements.
5. Parameter sensitivity check (Fig 6)

**Weaknesses:**

The paper frames the problem as "degree bias" (Figure 2) and attributes its success to fixing this. However, Negative Sampling in SGNS already have the mechanisms for mitigating degree bias (Residual2Vec). If (a) the primary issue is degree bias, (b) SGNS solves it, and (c) PSD only corrects for the degree bias, then the "FAKER w/o PSD" variant (which is essentially walks + SGNS) should perform similarly to the full FAKER model. The fact that it is not the case means that either (a), (b), (c) is false. Given  dramatically outperforms the "FAKER w/o PSD" variant strongly suggests the PSD mechanism is correcting for something more complex or different than just the first-order degree distribution that SGNS already handles, such as the degree assortativity.

The paper's core claim is that it creates a "balanced corpus," but it never provides evidence of this. A crucial piece of analysis is missing. The paper should have included a "post-correction" version of Figure 2. This would verify that the walk-visit shares for hubs and leaves actually did move closer to their true node shares after the PSD correction was applied. Without this, it is impossible to confirm that the degree bias was, in fact, mitigated.

It is also unknown if the correction may have introduced new biases. The performance gain is clearly from the PSD mechanism, but what it is truly capturing is left unexplored. Given that the PSD diagnosis focuses on "persistence" and "switching," it is highly plausible that it is capturing higher-order topological properties. For instance, Persistence (Z_LF) in a hub group is a proxy for high degree-assortativity, andSwitching (Z_HF) between hub and leaf groups is a proxy for degree-disassortativity. The performance gain may be from balancing these more complex structural transitions, not purely from fixing the node-degree distribution. The paper's explanation feels like an oversimplification.

**Questions:**

1. Could the authors provide an analysis of the final, corrected walk corpus (W_final)? Specifically, a reproduction of Figure 2 showing the post-PSD-correction visit shares for hubs and leaves? This would be critical to verifying the central claim that the degree bias was mitigated.
2. The paper attributes the performance gain to degree debiasing, yet the SGNS model already handles this. Have the authors considered that the PSD mechanism is capturing more complex, higher-order biases? For example, could Z_LF and Z_HF be acting as proxies for balancing degree-assortativity?
3. Following Q1, did the authors analyze whether the adaptive sampling, while correcting for persistence/switching, might have inadvertently introduced a new, different sampling bias (e.g., skewing the representation of "other" nodes)?
4. In the ablation study, how does the "FAKER w/o PSD" variant compare directly against a standard node2vec (or DeepWalk) + SI baseline? This would help isolate the contribution of the SI component from the PSD mechanism.

---

### Note · Authors · 2025-11-21

I have read and agree with the venue's withdrawal policy on behalf of myself and my co-authors.